# Isolation of *Leptospira interrogans* Serovar Canicola in a Vaccinated Dog without Clinical Symptoms

**DOI:** 10.3390/pathogens11040406

**Published:** 2022-03-27

**Authors:** Ivana Piredda, Sara Sechi, Raffaella Cocco, Loris Bertoldi, Bruna Palmas, Valentina Chisu

**Affiliations:** 1Istituto Zooprofilattico Sperimentale “G. Pegreffi” della Sardegna, Via Duca degli Abruzzi 8, 07100 Sassari, Italy; bruna.palmas@izs-sardegna.it (B.P.); valentina.chisu@izs-sardegna.it (V.C.); 2Teaching Veterinary Hospital, University of Sassari, Via Vienna 2, 07100 Sassari, Italy; sarasechilavoro@tiscali.it (S.S.); rafco@uniss.it (R.C.); 3BMR Genomics s.r.l., Via Redipuglia 22, 35131 Padova, Italy; loris.bertoldi@bmr-genomics.it

**Keywords:** Canicola, canine, dog, *Leptospira*, reservoirs, zoonosis

## Abstract

More than one million cases of leptospirosis occur across the globe annually, resulting in about 59,000 deaths. Dogs are one of the most important reservoirs of *Leptospira* species and play an important role in transmitting the pathogen to humans. Many of these infections are controlled by routine vaccination that has reduced the possible reintroduction of leptospiral serovars into the human population. However, it is still not clear how a vaccinated dog can become infected with one or more *Leptospira* serovars contained in the vaccine formulation and thus against which it should be immunized. Here, we present the case of an asymptomatic dog who developed leptospiral infection despite being vaccinated. This unusual case emphasizes the substantial impact of immunization on mitigating the acute signs of the disease, even while providing limited protection against infection. Further studies will be required to better understand the role of dogs in the environmental circulation of leptospiral serovars in Sardinia. Asymptomatic leptospiral infection in vaccinated dogs should be considered to allow for better diagnosis and management of the infection. This will be essential for preventing *Leptospira* outbreaks in the future.

## 1. Introduction

Canine leptospirosis is a zoonotic disease caused by spirochetes of the genus *Leptospira*. Although serovars Canicola and Icterohaemorrhagiae have been historically associated with canine leptospirosis in Europe [1], other serovars, such as Grippotyphosa, Australis, Hardjo, and Bratislava, have been detected in European dogs [2,3,4]. Because leptospiral infection is frequently asymptomatic in dogs [5,6], through a combination of being undiagnosed or diagnosed retrospectively, the disease is probably underdiagnosed, likely leading to a significant underestimation of the true burden of the disease. Vaccination is the most effective measure to control the spread of the infection in dogs and to prevent the development of leptospiral infection in humans [7,8]. In Europe, canine *Leptospira* vaccines containing antigens from different serogroups have been available for more than 60 years [9]. Currently, the vaccine commercially available in Italy is Nobivac L4 (MSD Animal Health), which is composed of four *Leptospira interrogans* strains, representing the serogroups Canicola, Icterohaemorrhagiae, Australis, and Grippotyphosa (http://www.ema.europa.eu; accessed on 28 December 2021). This inactivated bacterial vaccine has been shown to work well in dogs and to play a significant role in the development of protective immunity against leptospirosis [10]. However, cases of vaccinated dogs infected by *Leptospira* serogroups included in the vaccine formulation have been previously reported [11], highlighting that vaccination protects animals from clinical disease but may not prevent the infection if the animal is exposed to high bacterial load [12]. Leptospirosis should be considered as a potential cause of granulomatous hepatitis, even in vaccinated dogs and dogs without evidence of renal dysfunction or seroconversion [13]. Previous results from a serological study conducted in Sardinia indicated the presence of pathogenic *Leptospira* serotypes, including *L. interrogans* serovar Canicola, in vaccinated dogs [14]. In the same study, Piredda et al. speculated on the role of dogs in the possible transmission of *Leptospira* serovars to humans and on dogs’ contribution to make this region a zoonotic hotspot of the disease.

In this report, we present a case of canine leptospirosis documented by complete clinical presentation, diagnostic findings, and isolation of the etiological agent from the urine of one vaccinated dog from Sardinia, Italy. 

## 2. Case Report

A 7-year-old, sterilized female, Cirneco dell’Etna dog, weighing 18 kg, was presented to the Clinical Hospital of the University of Sassari for a regular routine check-up in April 2020. The dog, used in pet therapy, underwent clinical examination and blood analysis once a year to check overall health. According to the WSAVA guidelines [15], routine vaccinations (including Novibac L4 against canine leptospirosis) were up to date. Heartworm prophylactic treatment based on ivermectin (6 µg/kg) and pyrantel pamoate (5 mg/kg) was also given orally once a month.

The dog appeared well, and physical examination showed no abnormalities. Lymph nodes were normal in shape and of average size. Temperature (38.5 °C), respiratory rate (20 breaths per minute), heart rate (80 beats per minute), and capillary refill time (<2 s) were normal. Palpation of the abdomen did not reveal any abnormalities. The dog was well-hydrated, and membranes were pink and moist. Appetite was normal and weight stable. Stools were of normal consistency and volume. 

Hematology and chemistry were performed at day 0 using an automated hematology system (Dimension RxL Max Integrated Chemistry System (Siemens Healthcare Diagnostics, Milan Italy)) and a chemistry analyzer (Dimension RXL, Siemens S.p.A., Milan, Italy), respectively. Electrophoresis of serum proteins was also performed using the GENIO S device (Interlab s.r.l., Rome, Italy). Biochemical analyses indicated a significant increase in liver enzymes with abnormal values of transaminases, consistent with liver dysfunction. Creatinine levels were also altered (Table 1), while no abnormalities were detected on complete blood count (CBC) and serum protein electrophoresis. The dog was then subjected to abdominal ultrasound, which was unremarkable. Based on these results, a specific supplement for liver problems (containing Silybum 160 mg, DL-methionine 60 mg, Cynara scolymus 50 mg, curcuma 50 mg, and phosphatidylcholine 30 mg) was given twice daily. A commercial dog food for liver disease was also utilized.

Seven days after the first sample, serum was collected from whole blood, and indirect immunofluorescent antibody (IFA) titers for *A. phagocytophilum*, *Ehrlichia canis*, *Leishmania infantum*, *Toxoplasma gondii*, *Rickettsia* spp., and *Bartonella* spp. were quantified. Microscopic agglutination test (MAT) for *Leptospira* was also performed at the seroimmunology laboratory (Istituto Zooprofilattico Sperimentale Sardegna, Sassari, Italy). The antibody titers were tested against eight different serogroups, as previously described [16]. All serological tests and the leptospirosis MAT were negative. At the same time, genomic DNA extraction and real-time PCR (qPCR) targeting the *lipL32* gene specific for pathogenic *Leptospira* from urine samples were performed, as previously described [14]. A total of 1000 μL of urine was suspended in EMJH-fluorouracil semisolid medium at 28 °C and cultured for a period of three months, as previously reported [17]. Amplification of the gene target specified above was achieved from the dog urine, while we failed to isolate the *Leptospira* strain from urine culture. 

A diagnosis of *Leptospira* was made based on the qPCR-positive result, and the dog was immediately treated with intramuscular penicillin (0.3 mg/10 kg once daily for 21 days). Hematology and chemistry were performed at days 7, 15, 30, 45, 60, and 90 following treatment initiation. Serum creatinine levels returned to normal at day 7 after starting the penicillin therapy, while glutamic pyruvic transaminase and alkaline phosphatase returned to within the reference interval by days 30 and 45 after initiation of antibiotics, respectively (Table 2).

Twenty-one days after the penicillin treatment, further blood and urine samples were collected due to the presence of leptospiruria and leptospiremia in the infected dog as determined by qPCR, MAT analyses, and urine-culture. The qPCR and MAT were negative. After approximately 40 days, positive urine-culture was obtained, and treatment with doxycycline (5 mg/kg orally twice daily for 21 days) was given.

Partial sequencing of the *rrs* gene of strain isolated from the dog’s urine resulted in a 100% identity match with analogous sequences of *Leptospira interrogans* serovar Canicola present in GenBank. Multi-locus sequence-typing (MLST) analysis was applied to the leptospiral DNA that had been isolated following the seven-loci scheme proposed by Boonsilp et al. [18], revealing sequence type 37 (ST = 37) belonging to *L. interrogans* serovar Canicola.

For whole-genome sequencing of the *Leptospira* strain that had been isolated from the dog’s urine, an Illumina library was prepared with the Illumina Nextera XT kit, following the manufacturer’s instructions, then checked with BioAnalyzer (Agilent, Santa Clara, CA, USA) and quantified using Qubit fluorometer (Thermo Fisher, Bedford, MA, USA). The library was finally pooled with other samples, loaded onto the MiSeq system (Illumina, Inc., Ann Arbor, MI, USA) and sequenced following the V3-300PE strategy, producing 655,676 paired-end reads. Raw reads were processed using Cutadapt (v1.16) [19] in order to remove short (length < 150 bp) and low-quality (q < 30) reads and residual adapter sequences. Cleaned reads were assembled using SPAdes (v3.12.0) [20] with standard parameters and assembly metrics were calculated using QUAST (v4.6.3) [21]. The estimated genome size was 4,377,069 bp, with a mean coverage of 32X, L50 of 140, and N50 of 9044. The assembled sequences are available at the NCBI database under accession number JAJSDI000000000. Finally, quantitative evaluation of genome assembly was done using Busco (v5.1.2) against the spirochaetia_odb10 lineage. In parallel, the MetaPhlAn 3 [22] was used with standard parameters to characterize the sequenced isolate at species level and to evaluate the presence of possible contaminants. Lineage analysis revealed complete overlap with the Spirochetia class (239/239 BUSCO marker), and this finding was also supported by MetaPhlAn, which detected no contamination. In addition, PhyloPhlAn 3 [23] was applied in accurate mode with low diversity for the large-scale phylogenetic profiling of 14 *Leptospira* genomes, including isolate 769341 and eight Canicola serogroup isolates, available in the NCBI database (accessed: 4 November 2021). Thus, the isolate 769341 was included within the pathogenic isolates belonging to *Leptospira interrogans* serovar Canicola serogroup. The final refined tree was drawn with iTOL [24] (Figure 1).

## 3. Discussion

Dogs can act as carriers for pathogenic *Leptospira* strains, maintaining the bacteria in their renal tubules and eliminating them through urine into the environment for prolonged periods after infection [7,25]. Proper management of chronically infected dogs should be implemented to reduce environmental contamination; however, the identification of such individuals remains challenging [26]. In this report, the case of one asymptomatic dog infected with *L. interrogans* Canicola, a serovar included in the vaccine formulation, has been documented. The dog from this case report had received the vaccine four months prior to disease detection (December 2019). Although the vaccines represent a key point in the control of canine leptospirosis and in preventing clinical disease and renal carrier status in about 85% of immunized dogs [27], vaccination failure has been described in previous studies [11]. Results from this study corroborate the hypothesis that vaccination does not always guarantee complete protection against *Leptospira* serovars, and that vaccines are designed to prevent the disease, but not the infection [28]. The presence of leptospiral DNA in vaccinated dogs has previously been reported on the island [29], raising important questions regarding the role of dogs in the epidemiology of leptospirosis. In the dog under study, the absence of clinical signs consistent with leptospiral infection, along with culture isolation of *Leptospira*, suggested that the infection was asymptomatic, and that the microorganism may remain in the body after recovery from acute infection. Vaccinated dogs could represent a zoonotic risk, as described by several authors [8,26,30,31,32]. Although the serovar Canicola detected in this study has been described in humans [33,34], the dog owners from this study tested negative for leptospiral infection. Moreover, even if anemia is common in dogs with leptospirosis [35], in this study we did not find significant changes in CBC of the infected dog. Transaminase levels were elevated, with increases of alkaline phosphatase (321 U/L), transaminase (GOT 125 U/L and GPT 555 U/L), and creatinine. (1.55 mg/dl). These results were in accordance with other studies in which the increase of liver values has been associated with infected dogs. In particular, the increase of aspartate aminotransferase–alanine aminotransferase levels were indicative of leptospiral infection [36]. Liver function typically returns to normal values after treatment of leptospirosis with antibiotics, as observed in this case report. 

In this study, leptospiral isolation from dog urine was reported, confirming that, although culture of leptospires is labor-intensive and isolation can take 8 to 12 weeks, isolation of a leptospiral strain can be achieved. Moreover, isolation of the bacterium in culture is essential to determine the phylogenetic behavior of the genus [37] and to specify the disease [38,39]. Further, our study showed that *Leptospira* was isolated from dog urine after penicillin treatment. Previous research has reported that penicillin is effective during the leptospiremic phase but was not always efficacious at eliminating leptospiral organisms from the renal tubules [40,41,42]. Since persistent leptospiruria occurs in dogs after penicillin treatment, a course of doxycycline is routinely used due to eliminate leptospiruria and prevent zoonotic transmission [43].

This case supports the fact that leptospirosis is a complex disease in terms of diagnosis, and clinicians should evaluate the possibility of persistent leptospiruria in dogs despite penicillin treatment. Moreover, vaccines should be regularly kept up to date for the presence of new serogroups and serovars in an area. In this study, MAT failed to detect the disease, but rt-PCR analysis and subsequent isolation from urine-culture confirmed its presence. This is in agreement with previous studies that have highlighted the need to use more than one diagnostic tool for the detection of *Leptospira* species in reservoir hosts [4,44].

## 4. Conclusions

Our results suggest that, although the advent of animal vaccines against specific serotypes has reduced the incidence of transmission to humans, clinicians should be alert to the risk of *Leptospira* infection in vaccinated dogs and to the possibility of transmitting infection to its owners. Future studies will focus on establishing the role of dogs in the zoonotic transmission of leptospirosis.

## Figures and Tables

**Figure 1 pathogens-11-00406-f001:**
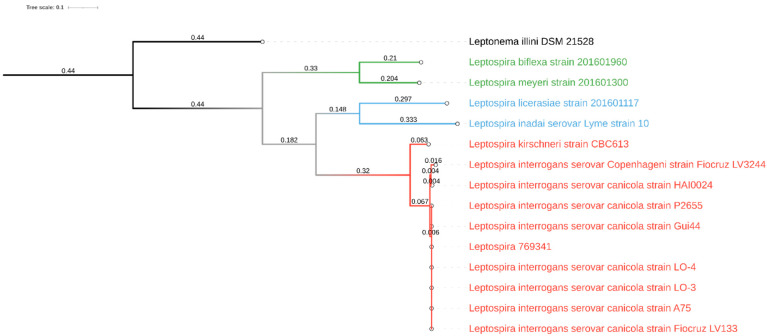
Graphical representation of the phylogenetic tree drawn with iTOL based on 14 whole-genome sequences of *Leptospira* spp. The various clades were colored differently according to their major group: pathogens in red, intermediates or opportunists in blue, and nonpathogens in green. *Leptonema illini* was used as the outgroup.

**Table 1 pathogens-11-00406-t001:** Metabolic results at day 0 indicating altered values of alkaline phosphatase, transaminase, and creatinine.

Parameter (Range)	Value
Albumin (2–3.3 gr/dL)	2.9
Alkaline phosphatase (1.5–90 U/L)	321
Total bilirubin (0.05–0.5 mg/dL)	0.2
Calcium (8–10 mg/dL)	10.7
Cholesterol (80–250 mg/dL)	210
Creatine phosphokinase (100–250 mg/dL)	128
Creatinine (0.5–1.5 mg/dL)	1.55
Gamma glutamyl transferase (6–16 U/L)	12
Glucose (50–100 mg/dL)	47
Glutamate oxaloacetate transaminase (25–72 U/L)	125
Glutamic pyruvic transaminase (30–85 U/L)	555
Phosphorus (3.5–6.5 mg/dL)	5.2
Total protein (5.3–8.3 g/dL)	6.8
Triglycerides (23–100 mg/dL)	114
Urea (20–50 mg/dL)	48

**Table 2 pathogens-11-00406-t002:** Metabolic profile in dog monitored at day 7, 15, 30, 45, 60, and 90 after the start of penicillin therapy.

Parameter (Range)	Day 7	Day 15	Day 30	Day 45	Day 60	Day 90
Alkaline phosphatase (1.5–90 U/L)	266	207	181	131	91	97
Total bilirubin (0.05–0.5 mg/dL)	0.05	0.01	0.01	0.01	0.01	0.3
Creatinine (0.5–1.5 mg/dL)	0.9	0.9	0.7	0.9	0.9	1
Gamma glutamyl transferase (6–16 U/L)	12	17	6	3	5	7
Glutamate oxaloacetate transaminase (25–72 U/L)	48	35	35	35	27	50
Glutamic pyruvic transaminase (30–85 U/L)	526	167	99	81	75	75
Urea (20–50 mg/dL)	15	12	11	9	16	22

## Data Availability

Not applicable.

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
