# Peer review of "Isolation of Leptospira interrogans Serovar Canicola in a Vaccinated Dog without Clinical Symptoms"

_pathogens, 2022, doi:10.3390/pathogens11040406_

Round 1

Reviewer 1 Report

In this case report, the authors described one regularly vaccinated dog that was infected with L. interrogans Canicola, a serovar included in the vaccine formulation. Results from this study corroborate the hypothesis that vaccination could not always guarantee complete protection against Leptospira serovars. The idea of describing such a case report is good because it can be helpful information for many clinicians. However, this case report doesn’t bring something new. Several authors have before described the failure of vaccination against pathogenic Leptospira spp. (serovars that are found in the vaccine).

  • G André-Fontaine,C Branger, A W Gray, H L B M Klaasen, Comparison of the efficacy of three commercial bacterins in preventing canine leptospirosis, Vet Rec 2003 Aug 9;153(6):165-9.

“Vaccination protects the animal against acute signs but may not prevent infection, if the dog is exposed to a high infectious challenge or to a highly invasive strain”

  • McCallum et al., Hepatic leptospiral infections in dogs without obvious renal involvement, J Vet Intern Med 2019 Jan;33(1):141-150

“In conclusion, leptospirosis should be considered as a potential cause of granulomatous hepatitis, even in vaccinated dogs and dogs without evidence of renal dysfunction or seroconversion”

Additional comments:

The English of this manuscript is poor. The whole manuscript should be revised.

Line 30: “have been recently detected in European dogs”

The references that were cited in the text are from 2013. For me, 2013 is not very recent. I think you should rephrase this sentence.

Line 31: “60% of dogs are asymptomatic”

Where did you find this information? In the references that you cited in the text, there is no such information.  Please correct it or put the citation properly.

Lines 38-39 “ L. Canicola, L. Icterohaemorrhagiae, L. Pomona, and L. Grippotyphosa”

On the product sheet of this vaccine is written something else. In addition, you wrote “L.Australis” instead of L.Pomona at lines 73-75. Which one is correct? I guess “L.Australis” since this is the correct information written on the vaccine’s product sheet.

Lines 50-51: “Our results, suggest that if vaccinated dogs become infected with serovar Canicola, they could be a source of leptospiral transmission to humans.”

 This sentence should be rephrased. The dogs can be a source of leptospiral transmission to humans regardless of serovar.

Lines 55-59: “… for a regular routine check-up which was carried out once a year due to certify the dog health state The dog underwent a clinical examination (including body condition, capillary reflux, temperature, mucosal color, etc.). EDTA blood and urine samples were aseptically collected by venopuncture and cystocentesis, respectively and then used for serological and molecular testing.”

 It’s hard to believe that you are doing cystocentesis as a routine analysis in an annual check visit. You mentioned that the blood was collected for serological and molecular testing. Why? Since this is just the annual check routine visit. I think the authors need to revise the whole part of the materials and methods.

Line 65: “The dog resulted well hydrated”

 The dog is not a “test” to “result” in something.

Ex. The dog was well hydrated or the dog showed no signs of dehydration.

Line 72: “The dog was regularly vaccinated (According to WSAVA)”

You should/must cite the WSAVA Guidelines and put it in the “References” section from the end.

 One of the most important pieces of information is missing. When was this dog last time vaccinated? Nothing is mentioned about it. What does it mean regularly? Please be more explicit. In my opinion, you cannot make the interpretation of MAT testing without having the information about the last vaccination. The MAT interpretation may be influenced if the dog was vaccinated recently.

Line 83:  You need to rephrase the name of the table.

Line 79: “The results of metabolic profile of blood serum resulted from the first blood collection  included a significant increase in the activity of liver enzymes and were consistent with worsening of the hepatic dysfunction.”

What does “first blood collection” mean? Does this first collection correspond with day0? Furthermore, you start to discuss about days 7,15,30, etc, but you didn’t explain what means the “first collection”. It was on the day when the dog was brought to the clinic?

Line 94: “Metabolic profile was performed at day 7 and 15 and then every 15 days thereafter (see Table 2).”

 The formulating of this sentence is wrong. If this is correct, where are the results for day 75? In the table are mentioned the following days: 7, 15, 30, 45, 60, 90. What about day 75?

Line 85: “A specific supplement for hepatic problems based on milk thistle, artichoke, turmeric, DL-methionine, and phosphatidylcholine, was given on an empty stomach to the dog and dosed at 2 tablets to take once daily. A commercial dog food for liver disease was also utilized.”

 When was this treatment started? Immediately after the “first blood collection” or a few days after? All these details are important. What is the dose for the hepatic supplement? 2 tablets is not a dose.

Line 99: “Culture isolation of Leptospira spp. from blood and urine samples was attempted.”

How? You mentioned something at line 109, but only for urine.

 Line 109: “A total of 1000 μL of urine was also suspended in EMJH-fluorouracil semisolid medium at 28°C and cultured for a period of three months.”

 What about blood cultivation?

 Line 115: “Once the diagnosis of leptospirosis was established, the dog was treated with intra-muscular penicillin at a dose of 0.3 ml/10kg per day for 21 days.”

 The dose for Penicillin is measured in U/kg or mg/kg not in ml!

Lines 100-104: “Moreover, the microagglutination test (MAT) in order to detect antibodies to leptospiral-specific antigens in the dog's serum was also performed. Live 7- to 10-day-old cultured strains of Leptospira species belonging to nine different serogroups (most frequently found in the Mediterranean area) were used as previously described [11]. Samples were considered positive when 50% or more of the leptospires were agglutinated at the 10-2  dilution, considering a cut-off titer of 100. The reactive serotype was considered to be the one with the highest titer.”

Which serovar had the highest titer?

References:

Barr, SC et al. (2005) Serologic responses of dogs given a commercial vaccine against Leptospira interrogans serovar Pomona and Leptospira kirschneri serovar Grippotyphosa. Am J Vet Res 66:1780–4. 34

Martin, LE et al. (2014) Vaccine-associated Leptospira antibodies in client-owned dogs. J Vet Intern Med 28:789–92.

“Post-vaccinal titers can be quite high (>1:6,400) and persist for months. The serovar with the highest titer cannot be assumed to be the infecting serovar, and it is typical to see an increase in several serovars”.

Since the last vaccination date is not mentioned, I think the interpretation of MAT can be problematic.

Line 115: “Once the diagnosis of leptospirosis was established, the dog was treated with intra- muscular penicillin at a dose of 0.3 ml/10kg per day for 21 days. After this, a treatment with tetracycline (10mg/kg once daily) was administered for 28 days.”

Can you provide a reference where is mentioned that a dog with leptospirosis should receive antimicrobial treatment for almost 2 months? When the treatment with antibiotics was started? How many days after the diagnosis?

119: “After 30 days, MAT serologic testing was repeated giving negative results.”

This is very hard to believe. Can you provide information regarding the results of MAT? You need to offer more details about the results using this test since it is considered as the gold standard method.

Line 122: “MLST”

Because you use this abbreviation for the first time, you need to explain it.

Author Response

Response to Reviewer 1 Comments

Comments and Suggestions for Authors

In this case report, the authors described one regularly vaccinated dog that was infected with L. interrogans Canicola, a serovar included in the vaccine formulation. Results from this study corroborate the hypothesis that vaccination could not always guarantee complete protection against Leptospira serovars. The idea of describing such a case report is good because it can be helpful information for many clinicians. However, this case report doesn’t bring something new. Several authors have before described the failure of vaccination against pathogenic Leptospira spp. (serovars that are found in the vaccine).

G André-Fontaine,C Branger, A W Gray, H L B M Klaasen, Comparison of the efficacy of three commercial bacterins in preventing canine leptospirosis, Vet Rec 2003 Aug 9;153(6):165-9.

“Vaccination protects the animal against acute signs but may not prevent infection, if the dog is exposed to a high infectious challenge or to a highly invasive strain”

McCallum et al., Hepatic leptospiral infections in dogs without obvious renal involvement, J Vet Intern Med 2019 Jan;33(1):141-150

“In conclusion, leptospirosis should be considered as a potential cause of granulomatous hepatitis, even in vaccinated dogs and dogs without evidence of renal dysfunction or seroconversion”

Additional comments:

The English of this manuscript is poor. The whole manuscript should be revised.

Response: The English has been revised.

Line 30: “have been recently detected in European dogs”

The references that were cited in the text are from 2013. For me, 2013 is not very recent. I think you should rephrase this sentence.

Response: This information has been rephrased in the new version.

Line 31: “60% of dogs are asymptomatic”

Where did you find this information? In the references that you cited in the text, there is no such information.  Please correct it or put the citation properly.

Response: The references have been corrected.

Lines 38-39 “ L. Canicola, L. Icterohaemorrhagiae, L. Pomona, and L. Grippotyphosa”

On the product sheet of this vaccine is written something else. In addition, you wrote “L.Australis” instead of L.Pomona at lines 73-75. Which one is correct? I guess “L.Australis” since this is the correct information written on the vaccine’s product sheet.

Response: The serogroup has been corrected.

Lines 50-51: “Our results, suggest that if vaccinated dogs become infected with serovar Canicola, they could be a source of leptospiral transmission to humans.”

This sentence should be rephrased. The dogs can be a source of leptospiral transmission to humans regardless of serovar.

Response: The sentence has been rephrased as suggested.

Lines 55-59: “… for a regular routine check-up which was carried out once a year due to certify the dog health state The dog underwent a clinical examination (including body condition, capillary reflux, temperature, mucosal color, etc.). EDTA blood and urine samples were aseptically collected by venopuncture and cystocentesis, respectively and then used for serological and molecular testing.”

It’s hard to believe that you are doing cystocentesis as a routine analysis in an annual check visit. You mentioned that the blood was collected for serological and molecular testing. Why? Since this is just the annual check routine visit. I think the authors need to revise the whole part of the materials and methods.

Response: The part of the materials and methods have been improved.

Line 65: “The dog resulted well hydrated”

The dog is not a “test” to “result” in something.

Ex. The dog was well hydrated or the dog showed no signs of dehydration.

Response: The sentence has been rephrased as suggested.

Line 72: “The dog was regularly vaccinated (According to WSAVA)”

You should/must cite the WSAVA Guidelines and put it in the “References” section from the end.

One of the most important pieces of information is missing. When was this dog last time vaccinated? Nothing is mentioned about it. What does it mean regularly? Please be more explicit. In my opinion, you cannot make the interpretation of MAT testing without having the information about the last vaccination. The MAT interpretation may be influenced if the dog was vaccinated recently.

Response: The manuscript has been improved, and I think most of our comments have

been addressed with either new analysis or necessary discussions.

Line 83:  You need to rephrase the name of the table.

Response: The name of the table has been rephrased.

Line 79: “The results of metabolic profile of blood serum resulted from the first blood collection  included a significant increase in the activity of liver enzymes and were consistent with worsening of the hepatic dysfunction.”

What does “first blood collection” mean? Does this first collection correspond with day0? Furthermore, you start to discuss about days 7,15,30, etc, but you didn’t explain what means the “first collection”. It was on the day when the dog was brought to the clinic?

Response: We revised the sentence and comments have been addressed with either new analysis or necessary discussions.

Line 94: “Metabolic profile was performed at day 7 and 15 and then every 15 days thereafter (see Table 2).”

The formulating of this sentence is wrong. If this is correct, where are the results for day 75? In the table are mentioned the following days: 7, 15, 30, 45, 60, 90. What about day 75?

Response: Infect this sentence lacks clarity and it has been modified

Line 85: “A specific supplement for hepatic problems based on milk thistle, artichoke, turmeric, DL-methionine, and phosphatidylcholine, was given on an empty stomach to the dog and dosed at 2 tablets to take once daily. A commercial dog food for liver disease was also utilized.”

When was this treatment started? Immediately after the “first blood collection” or a few days after? All these details are important. What is the dose for the hepatic supplement? 2 tablets is not a dose.

Response: All information about the integrator has been extensively described and added in the manuscript.

Line 99: “Culture isolation of Leptospira spp. from blood and urine samples was attempted.”

How? You mentioned something at line 109, but only for urine.

Response: The sentence has been rephrased and correct.

Line 109: “A total of 1000 μL of urine was also suspended in EMJH-fluorouracil semisolid medium at 28°C and cultured for a period of three months.”

What about blood cultivation?

Response: See response above.

Line 115: “Once the diagnosis of leptospirosis was established, the dog was treated with intra-muscular penicillin at a dose of 0.3 ml/10kg per day for 21 days.”

The dose for Penicillin is measured in U/kg or mg/kg not in ml!

Response: The unit of measurement has been corrected.

Lines 100-104: “Moreover, the microagglutination test (MAT) in order to detect antibodies to leptospiral-specific antigens in the dog's serum was also performed. Live 7- to 10-day-old cultured strains of Leptospira species belonging to nine different serogroups (most frequently found in the Mediterranean area) were used as previously described [11]. Samples were considered positive when 50% or more of the leptospires were agglutinated at the 10-2  dilution, considering a cut-off titer of 100. The reactive serotype was considered to be the one with the highest titer.”

Which serovar had the highest titer?

References:

Barr, SC et al. (2005) Serologic responses of dogs given a commercial vaccine against Leptospira interrogans serovar Pomona and Leptospira kirschneri serovar Grippotyphosa. Am J Vet Res 66:1780–4. 34

Martin, LE et al. (2014) Vaccine-associated Leptospira antibodies in client-owned dogs. J Vet Intern Med 28:789–92.

“Post-vaccinal titers can be quite high (>1:6,400) and persist for months. The serovar with the highest titer cannot be assumed to be the infecting serovar, and it is typical to see an increase in several serovars”.

Since the last vaccination date is not mentioned, I think the interpretation of MAT can be problematic.

Response: as reported in the new reformulated section of “case report” we specify that MAT results were negative in all phases. Infect, even if many authors emphasize that recent vaccination could give positive MAT titers against the serogroups used, it was not the case as reported in the discussion

Line 115: “Once the diagnosis of leptospirosis was established, the dog was treated with intra- muscular penicillin at a dose of 0.3 ml/10kg per day for 21 days. After this, a treatment with tetracycline (10mg/kg once daily) was administered for 28 days.”

Can you provide a reference where is mentioned that a dog with leptospirosis should receive antimicrobial treatment for almost 2 months? When the treatment with antibiotics was started? How many days after the diagnosis?

Response: this sentence was wrong and we reformulated it in the “case report” section.

119: “After 30 days, MAT serologic testing was repeated giving negative results.”

This is very hard to believe. Can you provide information regarding the results of MAT? You need to offer more details about the results using this test since it is considered as the gold standard method.

Response: please see above

Line 122: “MLST”

Because you use this abbreviation for the first time, you need to explain it.

Response: Done. 

Reviewer 2 Report

The study presented for publication is a case review. A dog is being recertified and results indicate an infection with L. Canicola. The dog is vaccinated with a tetra-leptospira vaccine including L. Canicola. The authors conclude that vaccination may not be a "useful strategy" for reducing the risk of leptospirosis transmisison to humans, but do not present any alternative or give examples of measures for control. Furthermore, the authors fail to indicate what is the rate of leptospirosis in humans or whether or not the owners of the dog were tested. The vaccine prevents urinary shedding and clinical disease, not infection. No vaccine has a 100% efficacy under field conditions. Moreover, we know 60% of dogs seem to have asymptomatic leptospirosis. This reviewer feels the conclusions need review and apply to this one case and not include the extrapolation to a population or to a control program where the vaccine is used as one element of control among others. A discussion on how infective a vaccinated dog can be is also needed.

Author Response

Response to Reviewer 2 Comments

Comments and Suggestions for Authors

The study presented for publication is a case review. A dog is being recertified and results indicate an infection with L. Canicola. The dog is vaccinated with a tetra-leptospira vaccine including L. Canicola. The authors conclude that vaccination may not be a "useful strategy" for reducing the risk of leptospirosis transmisison to humans, but do not present any alternative or give examples of measures for control. Furthermore, the authors fail to indicate what is the rate of leptospirosis in humans or whether or not the owners of the dog were tested. The vaccine prevents urinary shedding and clinical disease, not infection. No vaccine has a 100% efficacy under field conditions. Moreover, we know 60% of dogs seem to have asymptomatic leptospirosis. This reviewer feels the conclusions need review and apply to this one case and not include the extrapolation to a population or to a control program where the vaccine is used as one element of control among others. A discussion on how infective a vaccinated dog can be is also needed.

Response: Thank you for your review. The manuscript has been improved in the discussion section, and I think most of our comments have been addressed with other new information that have enriched the manuscript.

Reviewer 3 Report

The work is well conducted and written. However, the authors should consider including a brief discussion of vaccine efficacy to avoid misinterpretation because the vaccine did offer protection against clinical disease and renal carrier status (about 85%). 

Author Response

Response to Reviewer 3 Comments

Comments and Suggestions for Authors

The work is well conducted and written. However, the authors should consider including a brief discussion of vaccine efficacy to avoid misinterpretation because the vaccine did offer protection against clinical disease and renal carrier status (about 85%).

Response: Thank you for your review. We revised the discussion by adding your precious comments.

Reviewer 4 Report

The case report describes a case of vaccinated asymptomatic dog with L. interrogans serovar Canicola which was positive for leptospirosis. The case report is well written, and the paper has a good flow, but with many typos and English errors. These should be corrected. At the end of the Discussion section, and frequently, the zoonotic potential of infected dogs is mentioned. However, it is not stated whether the owners of this particular dog were tested and what was their result.  That information could thus corroborate the Conclusion section regarding the possibility of transmission.

Other issues:

Ln 13 replace "popular" with "frequent" or other expression

Ln. 29 delete "new" or use either "new" or "more"

ln 32 delete "it suggests that"

Ln 37 delete "from" and replace with "for"

Ln 38, 39 italicize properly: L. canicola........

Ln 44 L. Interrogans

Ln 44 serovar

Ln 45 ....Canicola still circulates in dogs, increasing the potential route of infection to humans.

Ln 50 Our results suggest

Ln 51 could be a source

Ln 55 sich is carried out one a year. Delete the rest

Ln 56. delete "a" clinical ...

Ln 57 Delete EDTA . Blood and urine samples...

Ln 61 .... normal in shape, of average size, not painful, swollen nor warm.

Ln 65 ...polydipsia and sensory was well.

65. The dog was well hydrated....

Ln 69 Italy

Ln 69 The complete blood count and the electrophoresis.....

Ln 71 alterations

Ln 72 The dog...

Ln 86 ... was prescribed on an empty stomach and dosed at 2 pills once daily.

Ln 87 was also prescribed

Ln 89 the dog

Ln 92 italicize latin names

Ln 93 delete "detection"

99. 102 111 italicize Leptospira

Ln 124 italicize L. interrogans

Ln 134 sequences are available

Ln 182. L. canicola

Author Response

Response to Reviewer 4 Comments

Comments and Suggestions for Authors

The case report describes a case of vaccinated asymptomatic dog with L. interrogans serovar Canicola which was positive for leptospirosis. The case report is well written, and the paper has a good flow, but with many typos and English errors. These should be corrected. At the end of the Discussion section, and frequently, the zoonotic potential of infected dogs is mentioned. However, it is not stated whether the owners of this particular dog were tested and what was their result.  That information could thus corroborate the Conclusion section regarding the possibility of transmission.

Response: Thanks for your kind reminders. Your suggestions have been immensely helpful. We revised the discussion and conclusion as you suggested.

Other issues:

Ln 13 replace "popular" with "frequent" or other expression

Response: the sentence has been reworded as suggested

Ln. 29 delete "new" or use either "new" or "more"

Response: It has been corrected

ln 32 delete "it suggests that"

Response: It has been corrected

Ln 37 delete "from" and replace with "for"

Response: the sentence has been reworded as suggested

Ln 38, 39 italicize properly: L. canicola........

Response: It has been corrected

Ln 44 L. Interrogans

Response: It has been corrected

Ln 44 serovar

Response: It has been corrected

Ln 45 ....Canicola still circulates in dogs, increasing the potential route of infection to humans.

Response: the sentence has been reworded as suggested

Ln 50 Our results suggest

Response: It has been corrected

Ln 51 could be a source

Response: It has been corrected

Ln 55 sich is carried out one a year. Delete the rest

Response: the sentence has been reworded as suggested

Ln 56. delete "a" clinical ...

Response: It has been corrected

Ln 57 Delete EDTA . Blood and urine samples...

Response: It has been corrected

Ln 61 .... normal in shape, of average size, not painful, swollen nor warm.

Response: It has been corrected

Ln 65 ...polydipsia and sensory was well.

Response: It has been corrected

  1. The dog was well hydrated....

Response: It has been corrected

Ln 69 Italy

Response: Done.

Ln 69 The complete blood count and the electrophoresis.....

Response: It has been corrected

Ln 71 alterations

Response: It has been corrected

Ln 72 The dog...

Response: It has been corrected

Ln 86 ... was prescribed on an empty stomach and dosed at 2 pills once daily.

Response: the sentence has been reworded as suggested

Ln 87 was also prescribed

Response: It has been corrected

Ln 89 the dog

Response: Done.

Ln 92 italicize latin names

Response: It has been corrected

Ln 93 delete "detection"

Response: Done.

  1. 102 111 italicize Leptospira

Response:

Ln 124 italicize L. interrogans

Response: It has been corrected

Ln 134 sequences are available

Response: It has been corrected

Ln 182. L. canicola

Response: Done.

Reviewer 5 Report

The manuscript is highlighting one of the unusual case study which elaborates discussion on asymptomatic presentation of leptospirosis in vaccinated dog and zoonotic threat. The methodology is designed well to better understand subclinical and therapeutic aspects of leptspiral infection through laboratory investigation, molecular analysis and drug-response relationship of infection with Lesptospira canicola. The authors have raised a valid point that how a vaccinated dog can become infected with one or more Leptospira serovars contained in the vaccine formulation!  Leoptospirosis is epidemiologically complex in terms of diagnosis and vaccine development due to presence of serogroups and serovars. I am sure that it was inactivated lepto vaccine given to dog under investigation. Sometimes, locality specific leptospira strain may not be protected by vaccine strain which needs to be taken into consideration. The UTI due to leptospiral infection in vaccinated dog leads to carrier state which is a matter of public health concern. The authors have concluded that the vaccination could not represent a useful strategy for reducing the risk of Leptospira transmission and further studies will be required to better understand changing pattern of asymptomatic leptospiral infection in vaccinated dogs and amend diagnosis and treatment protocol accordingly.

Author Response

Response to Reviewer 5 Comments

Comments and Suggestions for Authors

The manuscript is highlighting one of the unusual case study which elaborates discussion on asymptomatic presentation of leptospirosis in vaccinated dog and zoonotic threat. The methodology is designed well to better understand subclinical and therapeutic aspects of leptspiral infection through laboratory investigation, molecular analysis and drug-response relationship of infection with Lesptospira canicola. The authors have raised a valid point that how a vaccinated dog can become infected with one or more Leptospira serovars contained in the vaccine formulation!  Leoptospirosis is epidemiologically complex in terms of diagnosis and vaccine development due to presence of serogroups and serovars. I am sure that it was inactivated lepto vaccine given to dog under investigation. Sometimes, locality specific leptospira strain may not be protected by vaccine strain which needs to be taken into consideration. The UTI due to leptospiral infection in vaccinated dog leads to carrier state which is a matter of public health concern. The authors have concluded that the vaccination could not represent a useful strategy for reducing the risk of Leptospira transmission and further studies will be required to better understand changing pattern of asymptomatic leptospiral infection in vaccinated dogs and amend diagnosis and treatment protocol accordingly.

Response: Thank you for your review. 

Round 2

Reviewer 1 Report

I want to thank the authors for taking into account all my previous comments.  The authors replied to all the questions that I have raised during my first report. However, there are still some corrections to be made in the Materials and Methods section + discussion section.

Lines 89-95 "All serological tests as well as the leptospirosis MAT resulted negative.  Also, genomic DNA extraction and real time PCR (qPCR) targeting the lipL32 gene specific for pathogenic Leptospira from urine samples were performed as previously described [14]. A total of 1000 μL of urine was suspended in EMJH-fluorouracil semisolid medium at 28°C and cultured for a period of three months as previously reported [17]. After this period the urine-culture gave a negative result. A diagnosis of Leptospira was made and the dog was immediately treated with intramuscular penicillin (0.3 mg/10kg once daily, for 21 days)."

From this paragraph, I do not understand which method gave a positive result. MAT-negative; Urine culture - Negative; PCR - no details. However, the dog's diagnosis was Leptospirosis. Based on what results?

Lines 95-102: "A diagnosis of Leptospira was made and the dog was immediately treated with intra- muscular penicillin (0.3 mg/10kg once daily, for 21 days). Hematology and chemistry were performed at days 7, 15, 30, 45, 60, 90 following treatment initiation. Serum creatinine levels returned to normal at day 7 after starting the therapy, while glutamic pyruvic transaminase and alkaline phosphatase returned within the reference interval at days 30 and 45 after treatment, respectively (Table 2)."

When the therapy with antibiotics was started? For me, it is clear that on day 0 the hematology, chemistry, and ultrasound were performed, and specific liver supplements were prescribed.

However, it is not mentioned exactly when the diagnosis of Leptospirosis was made. All the methods: MAT, PCR (for leptospirosis), IFA (for other pathogens) were performed all on the same day (day 0) alongside hematology and chemistry??? Please be more specific.  You mentioned that creatinine levels returned to normal at day 7 after starting the therapy. Which therapy?  Supplements for liver, diet and the Penicillin were administered in the same day all of them?

3) "At day 21 after treatment initiation, the urine was collected again in order to evaluate Leptospira presence. The rt-PCR was negative and the isolation was attempted. Positive urine culture was obtained after approximately 40 days, and treatment with doxycycline (5 mg/kg orally twice daily for 21 days) was given."

I do not understand why the PCR and urine culture were retested, but not MAT? In such cases MAT is strongly recommended. It is well known that antibiotics are usually very effective in treating leptospirosis, and most dogs respond very quickly once antibiotics started. Penicillin is one of them. In general, PCR and urine culture are NOT recommended in dogs suspected of Leptospirosis that were treated with antibiotics before.

"Culture and PCR detect pathogenic leptospires or their nucleic acid, respectively, and have potential utility early in the course of untreated infection when antibody assays are frequently negative and antimicrobials have not yet been administered."

"Sykes J., 2011, "2010 ACVIM Small Animal Consensus Statement on Leptospirosis: Diagnosis, Epidemiology, Treatment, and Prevention", J Vet Intern Med, 25(1):1-13

“Although culture of leptospires is highly specific, sensitivity is quite low, which may be related to low-level bacteremia, administration of antibiotics prior to specimen collection, or technical difficulties associated with the assay.”

Reagan K. et al., 2019, Diagnosis of canine leptospirosis, Vet Clin North Am Small Anim Pract, Jul;49(4):719-731.

and many more.

4) Line 176: "Moreover, the initial antibiotic treatment has allowed the isolation of L. Canicola in culture,"

From this sentence I understand that antibiotics contributed to the isolation of L.Canicola in culture. This is false. Actually, antibiotic treatment makes isolation much more difficult.

In the discussions you need to include a special part to explain why you think the urine culture was positive despite the fact that it is known that in most cases the urine culture is negative after treatment with antibiotics.

Author Response

Response to Reviewer 1 Comments

Comments and Suggestions for Authors

I want to thank the authors for taking into account all my previous comments.  The authors replied to all the questions that I have raised during my first report. However, there are still some corrections to be made in the Materials and Methods section + discussion section.

Additional comments:

Lines 89-95 "All serological tests as well as the leptospirosis MAT resulted negative.  Also, genomic DNA extraction and real time PCR (qPCR) targeting the lipL32 gene specific for pathogenic Leptospira from urine samples were performed as previously described [14]. A total of 1000 μL of urine was suspended in EMJH-fluorouracil semisolid medium at 28°C and cultured for a period of three months as previously reported [17]. After this period the urine-culture gave a negative result. A diagnosis of Leptospira was made and the dog was immediately treated with intramuscular penicillin (0.3 mg/10kg once daily, for 21 days)."

From this paragraph, I do not understand which method gave a positive result. MAT-negative; Urine culture - Negative; PCR - no details. However, the dog's diagnosis was Leptospirosis. Based on what results?

Response: This information was missed in the previous version of the manuscript. Now it contains all requested data.

Lines 95-102: "A diagnosis of Leptospira was made and the dog was immediately treated with intra- muscular penicillin (0.3 mg/10kg once daily, for 21 days). Hematology and chemistry were performed at days 7, 15, 30, 45, 60, 90 following treatment initiation. Serum creatinine levels returned to normal at day 7 after starting the therapy, while glutamic pyruvic transaminase and alkaline phosphatase returned within the reference interval at days 30 and 45 after treatment, respectively (Table 2)."

When the therapy with antibiotics was started? For me, it is clear that on day 0 the hematology, chemistry, and ultrasound were performed, and specific liver supplements were prescribed.

Response: this information have been added in the text. Now, it has been explained that the penicillin treatment started after 7 days from the day 0.

However, it is not mentioned exactly when the diagnosis of Leptospirosis was made. All the methods: MAT, PCR (for leptospirosis), IFA (for other pathogens) were performed all on the same day (day 0) alongside hematology and chemistry??? Please be more specific.  You mentioned that creatinine levels returned to normal at day 7 after starting the therapy. Which therapy?  Supplements for liver, diet and the Penicillin were administered in the same day all of them?

Response: in this new version of the case report we have added the requested information

3) "At day 21 after treatment initiation, the urine was collected again in order to evaluate Leptospira presence. The rt-PCR was negative and the isolation was attempted. Positive urine culture was obtained after approximately 40 days, and treatment with doxycycline (5 mg/kg orally twice daily for 21 days) was given."

I do not understand why the PCR and urine culture were retested, but not MAT? In such cases MAT is strongly recommended. It is well known that antibiotics are usually very effective in treating leptospirosis, and most dogs respond very quickly once antibiotics started. Penicillin is one of them. In general, PCR and urine culture are NOT recommended in dogs suspected of Leptospirosis that were treated with antibiotics before.

"Culture and PCR detect pathogenic leptospires or their nucleic acid, respectively, and have potential utility early in the course of untreated infection when antibody assays are frequently negative and antimicrobials have not yet been administered."

"Sykes J., 2011, "2010 ACVIM Small Animal Consensus Statement on Leptospirosis: Diagnosis, Epidemiology, Treatment, and Prevention", J Vet Intern Med, 25(1):1-13

“Although culture of leptospires is highly specific, sensitivity is quite low, which may be related to low-level bacteremia, administration of antibiotics prior to specimen collection, or technical difficulties associated with the assay.”

Reagan K. et al., 2019, Diagnosis of canine leptospirosis, Vet Clin North Am Small Anim Pract, Jul;49(4):719-731.

and many more.

Response: I agree with the reviewer when he presented us some examples of studies in which the negative results after PCR and urine-culture have been obtained from dogs that received an antibiotic treatment. However, in the discussion we report our “unusual” result and discuss on the ineffective activity of the penicillin on the leptospiruria as supported by several authors.

4) Line 176: "Moreover, the initial antibiotic treatment has allowed the isolation of L. Canicola in culture,"

From this sentence I understand that antibiotics contributed to the isolation of L.Canicola in culture. This is false. Actually, antibiotic treatment makes isolation much more difficult.

In the discussions you need to include a special part to explain why you think the urine culture was positive despite the fact that it is known that in most cases the urine culture is negative after treatment with antibiotics.

Response: The sentence has been rephrased.

Round 3

Reviewer 1 Report

Dear authors,

Now the manuscript is much more concise and clear for all readers.

I want to thank the authors for taking into account all my previous comments. 

Best regards,